The toxicity assessment of phosmet on development, reproduction, and gene expression in Daphnia magna

Ataş Mustafa 1
http://orcid.org/0000-0001-9400-7892 Bereketoglu Ceyhun 2 3 ceyhun.bereketoglu@marmara.edu.tr
1 Managing Chemical, Biological, Radioactive, Nuclear Risks, Iskenderun Technical University , Hatay , Turkey
2 Department of Bioengineering, Marmara University , Istanbul , Turkey
3 Department of Biomedical Engineering, Iskenderun Technical University , Hatay , Turkey
Oehlmann Jörg
Electronic publication date: 2024 Feb 28
Publication date: 2024
Volume: 12
Electronic Location ID: e17034
Received 2023 Nov 14; Accepted 2024 Feb 8
Copyright: © 2024 Ataş and Bereketoglu
Copyright year: 2024
Copyright holder: Ataş and Bereketoglu
License: This is an open access article distributed under the terms of the Creative Commons Attribution License, which permits unrestricted use, distribution, reproduction and adaptation in any medium and for any purpose provided that it is properly attributed. For attribution, the original author(s), title, publication source (PeerJ) and either DOI or URL of the article must be cited.
License URL: https://creativecommons.org/licenses/by/4.0/

Keywords: Phosmet, Daphnia magna, Comet assay, Reproduction, Gene expression

Funding: Iskenderun Technical University and Marmara University, Turkey This study was supported by Iskenderun Technical University and Marmara University, Turkey. The funders had no role in study design, data collection and analysis, decision to publish, or preparation of the manuscript.

==============================
The use of pesticides to control pests, weeds, and diseases or to regulate plant growth is indispensable in agricultural production. However, the excessive use of these chemicals has led to significant concern about their potential negative impacts on health and the environment. Phosmet is one such pesticide that is commonly used on plants and animals against cold moth, aphids, mites, suckers, and fruit flies. Here, we investigated the effects of phosmet on a model organism, Daphnia magna using acute and chronic toxicity endpoints such as lethality, mobility, genotoxicity, reproduction, and gene expression. We performed survival experiments in six-well plates at seven different concentrations (0.01, 0.1, 1, 10, 25, 50, 100 μM) as well as the control in three replicates. We observed statistically significant mortality rates at 25 µM and above upon 24 h of exposure, and at 1 µM and above following 48 h of exposure. Genotoxicity analysis, reproduction assay and qPCR analysis were carried out at concentrations of 0.01 and 0.1 μM phosmet as these concentrations did not show any lethality. Comet assay showed that exposure to phosmet resulted in significant DNA damage in the cells. Interestingly, 0.1 μM phosmet produced more offspring per adult compared to the control group indicating a hormetic response. Gene expression profiles demonstrated several genes involved in different physiological pathways, including oxidative stress, detoxification, immune system, hypoxia and iron homeostasis. Taken together, our results indicate that phosmet has negative effects on Daphnia magna in a dose- and time-dependent manner and could also induce lethal and physiological toxicities to other aquatic organisms.

Introduction

Pesticides are commonly used agrochemicals to protect plants from pests such as insects, weeds, fungi, and bacteria, and increase agricultural productivity. Many of these chemicals are designed to disrupt the physiological activities of specific targets by causing dysfunction and reducing viability. However, the intensive use of pesticides on crops and seeds results in their leaching into the environment through various ways, which has become a major concern for both consumers and the environment (Ghasemzadeh, Sinaei & Bolouki, 2015). Several pesticides including organophosphates could induce toxicity at very low concentrations particularly on aquatic organisms (Karthick Rajan et al., 2023; Maggio, Janney & Jenkins, 2021; Maggio & Jenkins, 2022) Pesticides can enter the body of non-target organisms, including humans and animals, and pose health risks such as serious nervous system disorders, cancer, thyroid problems (Bilal et al., 2021; Buchanan et al., 2010; Tsaboula et al., 2016).

Organophosphates are a wide group of pesticides commonly used in agriculture and household applications. Phosmet is a non-systemic, phthalimide-derived organophosphate used to control moths, aphids, mites, suckers and fruit flies in plants and animals (Vasamsetti et al., 2021). Phosmet is a neurotoxic insecticide that acts by inhibiting acetylcholinesterase, an enzyme necessary for the transmission of nerve impulses (Kaur et al., 2022). Understanding the environmental fate and accumulation capability of phosmet is of critical importance to determine its possible adverse effects. According to the US Environmental Protection Agency (US EPA) 2010 report, the main exposure routes for phosmet are the runoff and entrainment of surface water. Phosmet has low solubility in water and according to the pesticide regulation database in 2010, the maximum phosmet concentration was reported to be up to 0.20 and 0.63 µg/L in groundwater and surface water, respectively (United States Environmental Protection Agency (US EPA), 2010). In the same report, it was indicated that peak model-estimated environmental concentrations in surface water resulting from different phosmet could be up to 245 µg/L (United States Environmental Protection Agency (US EPA), 2010). Phosmet has a low persistency and medium to low mobility with KFoc 482–757 mL/g in soil which results in high risk to soil-dwelling organisms (European Food Safety et al., 2021). Phosmet residues have also been detected in different sources including fruits, potato, olive oil and honey (El-Nahhal, 2020; Gomez-Ramos et al., 2020; Jara & Winter, 2019). Acute exposure to phosmet is highly toxic to freshwater fish and invertebrates, and chronic exposure has been shown to have adverse effects on the growth and survival of freshwater fish (United States Environmental Protection Agency (US EPA), 2010). Phosmet has been reported to be toxic to Daphnia magna (D. magna; 48 h EC50: ~0.01 mg/L), sunfish (96 h LC50: 0.07 mg/L), catfish (96 h LC50: 11 mg/L) and rainbow trout (96 h LC50: ~0.24 mg/L) (United States Environmental Protection Agency (US EPA), 2010). In a recent study on zebrafish, phosmet was observed to cause phenotypic abnormalities such as bradycardia, spinal curvature and growth retardation after 96 h of exposure (Vasamsetti et al., 2021). In the same study, transcriptomic analysis showed that phosmet affected different metabolic pathways including calcium signaling pathway, regulation of actin cytoskeleton, cardiac muscle contraction, and drug metabolism (Vasamsetti et al., 2021). In several studies on D. magna, it has been demonstrated that other organophosphate insecticides including malathion and chlorpyrifos cause mortality, developmental abnormalities, DNA damage as well as altering reproduction (Knapik & Ramsdorf, 2020; Palma et al., 2009a; Toumi et al., 2015).

D. magna is an aquatic model organism that has been widely used in toxicological analyses (Jordão et al., 2016; Shaw et al., 2008). As a model organism, D. magna offers several advantages, such as short life-cycle, small size and completed genome sequence (Ebert, 2005). Although phosmet has been reported to have some negative effects on different organisms, the chronic toxicity levels of these effects on reproduction have not yet been demonstrated. Moreover, organisms may respond to toxicants via different molecular mechanisms at transcriptional level. Determining acute and chronic effects and gene regulation profiles upon phosmet exposure is critical to establish relevant guidelines for toxicity tests and safety regulations. In this context, we aimed to determine the effects of phosmet on survival, reproduction, and gene expression levels on D. magna.

Materials and Methods

Chemicals

Phosmet was purchased from Sigma-Aldrich (purity > 99%). To prepare stock solutions, phosmet was dissolved in dimethylsulfoxide (DMSO; Sigma). The final concentration of DMSO in the assay was 0.1% (v/v).

Daphnia magna culture, maintenance and exposure

D. magna ephippia used were purchased from MicroBioTests Inc. (Daphtoxkit, Gent, Belgium). D. magna ephippia were first washed with tap water, then, using standard fresh water, they were incubated for 72–90 h at 20–22 °C and under continuous 6000 Lx illumination. The standard fresh water included 67.75 mg/L NaHCO3, 294 mg/L CaCl2, 123.25 mg/L MgSO4 and 5.75 mg/L KCl. Using an air pump, the standard fresh water was aerated for 30 min by bubbling air through a tube connected to an air pump. Neonates (<24 h) were fed with a suspension of Spirulina microalgae 2 h before being exposed to phosmet. For prolonged exposures, organisms were kept at a temperature of 22 ± 1 °C and a photoperiod cycle of 14/10 h light/dark and fed once daily with a mixture of Spirulina microalgae and yeast. The feeding ratios for Spirulina microalgae and yeast were adjusted to 1–2 × 105 and 1 × 105 cells/mL/day, respectively to ensure a stable and proper feeding. Half of the exposure water was changed on alternate days and concentration of the pesticide was kept constant throughout the experiments.

Survival analysis

For the survival assay, D. magna neonates (<24 h) were exposed to phosmet in 6-well plates with 10 organisms in each well in triplicates. Ten mL of standard fresh water containing 0.1% v/v DMSO (solvent control) or different concentrations of phosmet (0.01, 0.1, 1, 10, 25, 50, 100 µM) were used in the assay. Mortality rates were recorded at 24 and 48 h. When the organisms were examined under the microscope, they were considered dead if they were immobile and no movement was observed in their organs.

Comet assay

D. magna neonates (<24 h) were exposed to DMSO (solvent control), 0.01 or 0.1 μM phosmet. At 24 h, 20 organisms for each group were pooled in 1 mL of phosphate buffered saline (PBS; containing 20 mm EDTA and 10% DMSO) and homogenized according to alkaline Comet assay modified from Cavalcante, Martinez & Sofia (2008) to obtain a single cell suspension. The cell suspension was centrifuged in cold PBS (4 °C) at 10,000 rpm for 30 s and the pellet was retained. Single-cell gel electrophoresis was performed according to Singh et al. (1988). The slides were neutralized in ice-cold 0.4 M Tris buffer (pH 7.5), stained with 80 ml ethidium bromide (20 mg/mL), and examined at X400 magnification under a fluorescence microscope (Carl Zeiss Aksiostar Plus). For each sample, 100 cells were utilized and scored based on Collins (2004) to classify nucleoids according to tail formation. Assignment of nucleoids to one of five groups was as follows: 0 for DNA with no visible tail (no damage) and as four for almost all DNA in the tail (maximum damage). For comparison, the damage frequency (DF%), the arbitrary unit values (AU), and genetic damage index (GDI) were determined as defined by Pitarque et al. (1999) and Collins (2004). The experiments were performed in triplicate.

Chronic toxicity

For the chronic toxicity, D. magna neonates (<24 h) were used and exposure was carried out in 250 mL crystallization dish containing 100 mL of standard water with final concentrations of 0.1% v/v DMSO (solvent control), 0.01 or 0.1 μM phosmet. For each concentration, 10 organisms were used and the experiments were performed in triplicates. The reproduction assay was performed until all animals were dead and the number of offspring was recorded daily and removed from the dishes. The lifespan analysis was performed until all animals were dead. The number of live animals were recorded every day and dead organisms were removed from the dishes.

RNA extraction and gene expression analysis

RNA extraction was performed using the RNeasy Mini Kit (Cat. no.: 74104; Qiagen, Venlo, Netherlands) according to the mechanical lysis protocol provided by the manufacturer. For this, 20–25 D. magna neonates (<24 h) were exposed to 0.01, 0.1 μM phosmet or the solvent control. Following exposure, organisms were pooled together, flash-frozen in liquid, nitrogen and kept at −80 °C until further analysis. The RNA extraction was performed using the mechanical lysis protocol provided by the RNeasy Mini Kit (Qiagen, Hilden, Germany). The RNA concentrations were measured using a NanoDrop ND-100 UV-Vis spectrophotometer (Thermo Fisher Scientific Inc., Waltham, MA, USA) and cDNA was synthesized using qScript cDNA synthesis kit (Quanta Biosciences, USA) according to the manufacturer’s instructions. qPCR was carried out with a Rotor-Gene Q (Qiagen, Inc., Hilden, Germany), using a LightCycler FastStart DNA Master SYBR Green I Kit (Roche Molecular Biochemicals, Mannheim, Germany), according to the manufacturer’s instructions. The thermocycling conditions were as follows: one cycle of initial denaturation at 95 °C for 10 min, followed by 40 quantification cycles of denaturation at 95 °C for 10 s, primer annealing at primer-specific temperatures for 10 s, and primer extension at 72 °C for 25 s. The relative gene expressions were calculated according to the ΔΔCt method, as described by Schmittgen & Livak (2008). Normalization was performed using the reference gene actin (act1). The experiments were conducted with at least four replicates. Primer sequences for the analyzed genes were given in Table 1.

Table 1 The primer sequences used for qPCR.

Gene symbol	Gene name	Forward primer (5′–3′)	Reverse primer (5′–3′)	
mt-1	metallothionein 1	TTGCCAAAACAATTGCTCAT	CACCTCCAGTGGCACAAAT	
mt-a	metallothionein a	GAGCGCCATGCCAAAATCCC	TCGTCGTTGTAAAATCCGCCT	
mt-b	metallothionein b	TGGAACCGAATGCAAATGCG	CGGACTTGCATGGACAACTG	
mt-c	metallothionein c	AAAGTGTGCCCTCGTTGTCA	CTTACAGTCGTCCCCACACG	
cat	catalase	TGGCGGAGAAAGCGGTTCAGC	GTGCGTGGTCTCTGGGCGAA	
gst	glutathione S transferase	TCAGGCTGGTGTTGAGTTTG	GAGCAAGCATTTGTCCATCA	
dap1	death associated protein 1	ATGGCCTTGGCTGCCTCTGGA	GCGGGGGACGTTTGCCATTT	
hsp70	heat shock protein 70	CGACGGCGGGAGATACGCAC	CCACGGAAAAGGTCGGCGCA	
hsp90	heat shock protein 90	CCCTCTGTGACACTGGTATTGGCA	GCCCATGGGTTCTCCATGGTCAG	
NOS1	nitric oxide synthase 1	ACGCAACTCGGTGACAGCGG	AGGCGTGAGCGGCCAGTAGA	
NOS2	nitric oxide synthase 2	GGCACCCGCTTGTTGGCACT	GCGTGCCCCTCACTTGAGCC	
CYP4	cytochrome P450 4	AGCCGAGCACCAACAGCGAA	GCGGGCCGGTCAGAATCACC	
CYP314	cytochrome P450 314	TCTTGGGTCGGCGTCTGGGA	TCGCGGGTGTCAACGCCTTC	
hr96	nuclear hormone receptor 96	GCGGAGACAAGGCTTTAGGTT	AGGGCATTCCGTCTAAAGAAGGCT	
magro	magro	GCATAGGACGTGAGATGGTTAG	ACAAGAAGCTCGCATGGTTA	
NPC1b	Niemann Pick type C	TCATAGGTGGACAGCAAGATTAC	TAGCAGGCACACCAACATAG	
SM3	sphingomyelinase 3	GCGCTCTTCCAGCTCTATTT	GACGGATTTGCTCGCATTTG	
hif1a	hypoxia-inducible factor-1	GGTCCAGACCCAAGCAGCCAGGC	GTCCAGGAGCAGCAGCCAGC	
ftn3	ferritin-3	GGTGATGGCCTAGGAGTCTTT	TGCTCCAAACTTTAGATGCTTT	
man	mannosidase	GGTTCCCTGGAGTTTATGGTAG	AGTCGTCGGTGAATCTGTTG	
vtg1	vitellogenin-1	CCAGCGAATCCTACACCGTCAAG	GAGCCGCACAGACCACAGAG	

Statistical analysis

The statistical analyses were performed using GraphPad Prism 8 software (GraphPad Software, San Diego, CA, USA). Outliers were identified and excluded from the results. One-way ANOVA followed by Dunnett’s test was used for multiple comparisons. For 50% lethal concentration (LC50), nonlinear regression analysis was performed. For comet assay, the obtained data sets were first tested for normality (Shapiro–Wilk test) and homogeneity (Levene’s test) tests before statistical analysis and then subjected to one-way ANOVA. The difference was accepted significant when the p value was < 0.05 (*p < 0.05; **p < 0.01; ***p < 0.001; ****p < 0.0001).

Results

Acute toxicity in D. magna upon exposure to phosmet

Survival assay was performed to determine whether various concentrations of phosmet can cause acute toxicity. Our results showed that phosmet causes mortality in a dose- and time-dependent manner. We found that exposure of D. magna for 24 h did not show any lethality in response to 0.01 and 0.1 µM phosmet, while 1 µM phosmet decreased mobility in organisms. Mobility of organisms was significantly decreased, while movement was observed in their organs in response to 10 µM phosmet. However, we did not observe any significant lethality from 0.01 to 10 µM phosmet concentrations (Fig. 1A). We found a significant mortality (44.4%) in response to 25 µM phosmet and the living organisms in this group demonstrated less mobility and organ movement. Fifty and 100 µM doses resulted in higher mortality (50.0% and 71.1%, respectively) and the living organisms did not show any mobility but slight organ movement (Fig. 1A). Exposure of D. magna for 48 h for 1 and 10 µM doses resulted in significant mortality (63.3% and 88.9%, respectively) (Fig 1B), while these doses did not show this effect after 24 h. Exposure to 25 µM phosmet caused 87.8% mortality, while exposure to 50 and 100 µM phosmet for 48 h resulted in 100% mortality (Fig. 1B). We also determined LC50 of phosmet at 24 and 48 h. We found that the LC50 value was 36.87 µM at 24 h, while it was 0.81 µM at 48 h (Figs. 1C and 1D).

Figure 1 Phosmet resulted in mortality.

Daphina magna neonates (<24 h old) were exposed to phosmet (0.01, 0.1, 1, 10, 25, 50, 100 µM) in six-well plates with 10 organisms in each well in triplicates. Mortality rates were recorded at 24 h (A) and 48 h (B). LC50 of phosmet was determined at 24 h (C) and 48 h (D). Statistical analyses were performed using one-way ANOVA followed by Dunnett post-test and nonlinear regression. The difference was accepted significant if p values < 0.05. n = 3. Phosmet cause mortality in a dose- and time-dependent manner with a significant lethality starting with 25 and 1 µM at 24 and 48 h, respectively. *p < 0.05; ***p < 0.001; ****p < 0.0001.

Phosmet causes DNA damage in D. magna

The mean values and standard deviations of the DNA damage indicators, DF%, AU and GDI in D. magna upon exposure to 0.01 and 0.1 µM phosmet, and the control are summarized in Table 2. The results showed that increasing concentrations of phosmet caused significant DNA damage in D. magna compared to the control group. We observed that 0.01 and 0.1 µM phosmet resulted in 53.33% and 63.67% DF%, respectively, while the control group showed 37.33% DF%. Similarly, AU and GDI values increased in parallel with the phosmet concentrations and significantly differed from the control group. AU values were determined to be 75.00, 127.33, and 140.00 for the control, 0.01 and 0.1 µM phosmet, respectively. Meanwhile, GDI values were determined to be 0.75, 1.27, and 1.40 for the control, 0.01 and 0.1 µM phosmet, respectively (Table 2).

Table 2 DNA damage in Daphnia magna in response to phosmet.

	Damage frequency (%)	Arbitrary unit (AU)	Genetic damage index (GDI)	
Control	37.33 ± 0.88a	75.00 ± 6.65a	0.75 ± 0.06a	
0.01 µM	53.33 ± 0.87b	127.33 ± 3.17b	1.27 ± 0.03b	
0.1 µM	63.67 ± 0.81c	140.00 ± 2.64b	1.40 ± 0.02b	
p value	***	***	***	
Notes:

DF, Damage frequency; AU, Arbitrary Units; GDI, Genetic Damage Index (mean value ± sd). The differences between groups were considered statistically significant if the p value was <0.05.

*** p < 0.001

Groups with different letters are significantly different from each other.

The effects of phosmet on reproduction

In order to evaluate the chronic effects of phosmet on D. magna, we performed reproduction and lifespan analyses. As 0.01 and 0.1 µM phosmet concentrations did not show significant mortality in acute toxicity experiments, we considered these concentrations to be suitable for the physiological and gene expression analyses. We conducted reproduction and lifespan analyses until all Daphnids were dead during which the number of offspring and dead organisms were recorded daily, and removed from the dishes. The average of all replicates for the total number of offspring that one single Daphnia produced over the exposure period were calculated and statistically analyzed. The results indicated that 0.01 µM phosmet did not show any effect on reproduction, while 0.1 µM phosmet resulted in significantly higher offspring per adult compared to the control group (Fig. 2). The control and 0.01 µM exposure groups started producing progeny at 7th day, while 0.1 µM exposure group did not give offspring until day 24 (Fig. S2). Interestingly, although 0.1 µM phosmet started producing progeny very late compared to the other groups, the average total number of offspring per adult was significantly higher.

Figure 2 Phosmet increased reproduction.

Daphnia magna neonates (<24 h old) were exposed to phosmet (0.01 and 0.1 µM) and number of offspring was recorded daily until all Daphnids were dead. For each concentration, 10 organisms were used and experiments were performed in triplicates. The average of all replicates for the total number of individuals that one single Daphnia produced over the exposure period was taken. Statistical analyses were performed using one-way ANOVA followed by Dunnett post-test and the difference were accepted significant if p values < 0.05. n = 3.0.1 µM exposure group started to give offspring later than the control and 0.01 µM groups, however, the average number of offspring per adult was significantly higher. An asterisk (*) indicates p < 0.05.

The effects of phosmet on gene expression profiles

To determine the possible molecular mechanisms behind the phosmet toxicity, we further analyzed the expression levels of stress response genes following 24 h exposure to phosmet. The analyzed genes are involved in key pathways such as heat shock proteins, metal response, oxidative stress, immune and apoptotic pathway. We found that the metallothionein gene mt-1 was significantly downregulated in response to 0.01 µM phosmet (Fig. 3A). We found that an immune response gene, nitric oxide synthase 1 (NOS1) and stress response gene, gst were repressed by both 0.01 and 0.1 µM phosmet concentrations (Figs. 3B and 3C). We also determined the expression profiles of other stress biomarkers CYP genes. CYP4 was strongly repressed upon exposure to 0.01 and 0.1 µM (Fig. 3D), while CYP314 was downregulated by only 0.01 µM phosmet (Fig 3E). Interestingly, other stress response genes such as mt-a, mt-b, mt-c, hsp70, hsp90, catalase (cat), and death associated protein 1 (dap1) were not affected by any dose (Supplemental File 1).

Figure 3 Phosmet changed the expression of genes involved in stress response and xenobiotic metabolism.

Daphnia magna neonates (<24 h old) were exposed to 0.01 and 0.1 μM phosmet for 24 h, and the expression levels of mt-1 (A), NOS1 (B), gst (C), CYP4 (D), and CYP314 (E) were determined using qPCR. Statistical analyses were performed using one-way ANOVA followed by Dunnett post-test and the difference were accepted significant if p values < 0.05. n ≥ 4. All the genes were significantly downregulated by phosmet. *p < 0.05; **p < 0.01.

The expression of genes involved in lipid metabolism were also investigated. We found that nuclear hormone receptor 96 (hr96), and Niemann Pick type C (NPC1b) were significantly downregulated by both 0.01, 0.1 µM phosmet (Figs. 4A and 4C). Meanwhile, magro was repressed by only 0.01 µM phosmet (Fig. 4B), while sphingomyelinase 3 (SM3) was significantly downregulated in response to 0.1 µM group (Fig. 4D). Interestingly, another lipid metabolism related gene, mannosidase (man) did not show any expression changes by any concentration (Supplemental File 1). We also investigated the expression of a reproduction related gene, vitellogenin-1 (vtg1), and found that although vtg1 expression was induced, the increase was not significant (Supplemental File 1). To observe possible negative effects on respiration, we determined the expression of genes related to respiration and iron homeostasis. We observed a significant downregulation of hypoxia-inducible factor-1 (hif1) and ferritin-3 (ftn3) in response to both 0.01 and 0.1 µM (Figs. 4E and 4F).

Figure 4 Phosmet decreased the expression of genes associated with lipid metabolism and respiration.

D. magna neonates (<24 h old) were exposed to 0.01 and 0.1 μM phosmet for 24 h, and the expression levels of hr96 (A), magro (B), NPC1b (C), SM3 (D), hif1a (E), and ftn3 (F) were determined using qPCR. Statistical analyses were performed using one-way ANOVA followed by Dunnett post-test and the difference were accepted significant if p values < 0.05. *p < 0.05; **p < 0.01. n ≥ 4.

Discussion

The excessive and inappropriate use of organophosphates cause them to end up in different compartments of the environment including surface, groundwater, and soil (Bilal et al., 2021; Rahman et al., 2021). Understanding the impacts of these compounds including phosmet is of vital importance to take the necessary measurements by the regulatory organizations. Hence, in the present study, we investigated the impacts of phosmet on several parameters including survival, reproduction and gene expression to better understand its negative impacts.

Survival assay data demonstrated that phosmet administration leads mortality in a dose- and time-dependent manner, as evidenced by phosmet concentrations higher than 10 µM causing significant lethality in D. magna upon exposure to 24 h. At 48 h exposure, even 1 µM phosmet caused significant mortality, while high concentrations (50 and 100 µM) killed all the organisms. We also observed that the living organisms in lethal concentrations of phosmet show a decrease in organ movement and mobility. Similarly, it has been shown that phosmet induces developmental toxicity in a time- and dose-dependent manner, as well as an abnormal touch-evoked response and swimming indicating a teratogenic effect of phosmet on zebrafish embryos (Vasamsetti et al., 2020). In another study on zebrafish, it was indicated that phosmet caused phenotypic abnormalities such as bradycardia, spinal curvature and growth retardation after 96 h of exposure, and a concentration of 25.2 µM phosmet caused statistically significant mortality in all 24, 48, 72 and 96 h of exposure (Vasamsetti et al., 2021). Supporting our results, in an in vitro study, it was found that phosmet affected cellular viability in a concentration- and time-dependent manner (Guinazu et al., 2012). We further determined LC50 of 36.87 µM and 0.81 µM for phosmet at 24 and 48 h, respectively. It was previously shown that the effective dose (EC50) of phosmet on D. magna after 48 h of exposure was 0.018 µM (United States Environmental Protection Agency (US EPA), 2010). On zebrafish embryos, LC50 of phosmet has been determined to be 7.95 ± 0.30 mg/L (~25.05 µM) at 96 h (Vasamsetti et al., 2020). In an acute oral toxicity study, LC50 of phosmet has been found as ~77.2 and ~74.4 µM for Scaptotrigona bipunctata and Tetragonisca fiebrigi, respectively (Dorneles, de Souza Rosa & Blochtein, 2017). Altogether, we speculated that the time of exposure and the type of organism could change the effective level of phosmet and at higher concentrations, it could show teratogenic activity and lethal effect on organisms.

Environmentally relevant concentrations of a chemical may show a realistic exposure scenario to better reveal the long term effects at physiological and molecular level. In the present study, we performed genotoxicity, reproduction, and gene expression analyses with 0.01 and 0.1 µM phosmet, as these concentrations are relevant to its reported doses in different environmental compartments (United States Environmental Protection Agency (US EPA), 2010) and did not resulted in significant mortality in our acute toxicity experiments. Comet assay is an efficient technique used to detect DNA damage based on single strand breaks or structural DNA changes (Hilgert Jacobsen-Pereira et al., 2018). Several studies have used comet assay and shown that increase in DNA damage in D.magna was associated with exposure to several pesticides such as malathion (Knapik & Ramsdorf, 2020), triclosan and carbendazim (Silva et al., 2015, 2017, 2019). Other studies have also indicated that different compounds including 17α-ethinylestradiol (Rodrigues, Silva & Antunes, 2021), diclofenac, ibuprofen and naproxen (Gómez-Oliván et al., 2014) could induce single strand DNA breaks. In the present study, we demonstrated that even the lowest concentration of phosmet was able to promote significant DNA damage in D. magna cells. We also observed that increase in the concentration resulted in increase in the level of DNA damage. Although, it has been indicated that phosmet is unlikely to be genotoxic in vivo (Anastassiadou et al., 2021), its genotoxicity and mutagenic activity have been previously shown in Salmonella and Saccharomyces assays (Vlcková et al., 1993). Taken together, we reason that phosmet is genotoxic to D. magna and the observed DNA damage in the cells could be due to oxidative stress produced by ROS (Lee, Kim & Choi, 2009).

We further performed reproduction assay and interestingly, we observed a very low number offspring after 21 days. We speculated that this observation may be an evidence of suboptimal nutritional conditions which could potentially induce intraspecific variations in D. magna. Consequently, this lead us to analyze the reproduction parameter until all Daphnids were dead and we observed that the 0.1 µM phosmet group (the second lowest sub-lethal concentration) gave significantly higher offspring per adult compared to the control group. The hormetic response refers to a phenomenon where exposure to low levels of stressors can induce a response in an organism, resulting in improved resistance or adaptability (Calabrese & Baldwin, 2002). Similarly to our study, it has been shown that exposure to chlorpyrifos increased the offspring number per adult in Daphnia carinata compared to control which is an evidence of hormesis (Zalizniak & Nugegoda, 2006). Although hormesis has been recognized and reported for various toxicants over the time, there are limited reports on hormesis involving organophosphates (Stark & Vargas, 2003). We speculated that the hormetic effect of phosmet in our case could be due to induced intraspecific variations as well as molecular responses against oxidative stress and damage. However, the mechanism behind this remains unknown and requires further analyses. Because, it is essential to recognize that relying solely on the number of offspring is insufficient for drawing conclusions about sensitivity to a specific toxicant. We also found that 0.1 µM exposure group started to give offspring very late (24th day) compared to the control and 0.01 µM exposure groups (7th day). Reproduction parameters, such as brood size and body length of adults could be important indicators for the difference between phosmet exposure and the control. Previously, it has been shown that exposure to chlordane and endosulfan, organochlorine insecticides, resulted in significant decrease in the number of neonates, the mean brood size, and body size of adults as well as a delay in the first brood of D. magna (Manar, Bessi & Vasseur, 2009; Palma et al., 2009b). It is important to mention that the main drawback of this study is that we were unable to measure the body size of adults during the chronic toxicity analysis. Taken together, we suggest that even though individuals exposed to phosmet showed delayed reproduction, the higher number of offspring reported in the present study could be related to a larger body size of these individuals. However, to elucidate this, further experiments needs to be conducted. In contrast to our findings, previous studies on other organophosphates such as chlorpyrifos and malathion have shown a significant decrease in the number of neonates per adult upon exposure to pesticides (Palma et al., 2009a; Toumi et al., 2015).

Another critical point is that several pesticides may exert an endocrine disrupting effect on organisms particularly at low concentrations and additive interactions with other mechanisms (Petrelli & Mantovani, 2002). It has previously been shown that elevated ROS due to oxidative stress may cause deleterious effects on ovarian tissue (Shokoohi et al., 2019). We also suggested that exposure to phosmet may cause oxidative stress which in turn leads to production of free radicals and ROS, and this may cause a delay in reproduction. Taken together, we speculated that the negative effects of phosmet on reproduction could be species-specific and other molecular mechanisms might also be involved in response to phosmet.

Gene expression analysis is widely used as an efficient technique in determining the molecular mechanisms and possible biomarkers behind the toxicity of various chemicals including pesticides (Li et al., 2023; Salesa et al., 2022; Weil et al., 2009). Genetic biomarkers may be associated with either protective processes or induction of detrimental effects (Pradhan et al., 2020). Since we observed DNA damage and lethal effects of phosmet, in this study, we further analyzed the expression profiles of several genes involved in stress response pathways. Metallothioneins are small and cysteine-rich heavy metal-binding proteins that are found in many organisms from invertebrates to mammals and are associated with critical biological roles such as metal homeostasis and detoxification, protection against oxidative stress and heavy metals, neuroprotection, and anti-inflammatory mediators (Amiard et al., 2006; Pradhan et al., 2020; Rodríguez-Menéndez et al., 2018; Ruttkay-Nedecky et al., 2013). Several studies have demonstrated that decreased expression of metallothionein genes increase vulnerability of organisms to different conditions including oxidative stress, metal toxicity and metabolic disorders (Huang et al., 2021; Sato et al., 2010; Wu et al., 2019). Hence, in the present study, repressed expression of mt-1 and NOS1 could be an indicator to the observed lethality, physiological impacts, and neurotoxicity of phosmet to D. magna, as downregulation of these genes could make D. magna more susceptible to environmental stressors.

Organisms have evolved sophisticated detoxification systems to overcome the adverse effects of oxidative stress. cytochrome P450 (CYP) enzymes play role in the phase I detoxification pathway, while gst encodes the phase II xenobiotic metabolizing enzyme GST that uses the products of phase I reactions in formation of large endogenous molecules subsequently eliminating toxic xenobiotics (Benson & Di Giulio, 2018; Gunderson et al., 2018; Mukhopadhyay & Chattopadhyay, 2014). In the present study, downregulation of gst in response to phosmet indicated a decrease in capability of D. magna to excrete phosmet from the cells. Similarly, several other studies have also demonstrated downregulation of gst in response to different compounds in various organisms (Pradhan et al., 2020; Seyoum et al., 2021; Wang et al., 2018). We determined a significantly repressed expression of CYP4 and CYP314 upon exposure to phosmet. Of these, CYP4 is associated with metabolizing fatty acids and steroids and is particularly used as indicator in insecticide resistance (Jarrar & Lee, 2019; Scharf et al., 2001), while CYP314 is involved in the conversion of ecdysone to its active form (Shen et al., 2003). In another study performed on D. magna, it has been shown that CYP4 and CYP314 were significantly downregulated in response to other pesticides, glyphosate and methidathion, respectively (Le et al., 2010). Taken together, we speculated that exposure to phosmet could cause oxidative stress which subsequently negatively affected fatty acid and steroid metabolisms in D. magna (see subsequent parts). Although, we did not observe any significant change in other stress response genes including catalase (cat), several biochemical oxidative stress indicators including superoxide dismutase (SOD), glutathione peroxidase (GPx), glutathione (GSH), and CAT have been previously shown to be induced in response to phosmet in rainbow trout (Muhammed & Dogan, 2021). We suggest that further studies to measure the levels of such biomarkers could reveal the negative impacts of phosmet on D. magna.

Lipid molecules play an essential role in signaling, membrane composition, and energy production (Anderson, Cole & Williams, 2004). Dysfunction of lipid metabolism may lead to several problems including obesity, diabetes, and atherosclerosis (Joffe, Panz & Raal, 2001; McNeely et al., 2001; Watanabe et al., 2008). In the present study, we observed significant downregulation of all the analyzed lipid related genes by phosmet exposure except man which did not show significant expression change. Of these, hr96 is involved in controlling energy metabolism through homeostasis and transport of triacylglycerols and cholesterol by regulating several genes including NPC1b, SM3, magro and man (Sengupta et al., 2017; Sieber & Thummel, 2012). NPC1b also plays role in cholesterol and fatty acid homeostasis. Sphingomyelins is a type of lipid that is highly abundant in neonates and is essential for the maturation of D. magna (Sengupta et al., 2017). magro is a key biomarker of hr96 activation that plays critical role in hydrolysis of cholesterol esters and stimulation of cholesterol clearance from the intestine of Drosophila melanogaster (Sengupta et al., 2017; Sieber & Thummel, 2012). Downregulation of these genes suggests that exposure to phosmet may inhibits fatty acid uptake in D. magna and this subsequently may result in lipid accumulation in the cells. However, further analysis is needed to demonstrate whether cells accumulate lipids. Other studies on D. magna have also demonstrated that exposure to various environmental pollutants including plasticizers and per- and polyfluorinated alkyl substances could cause such adverse effects (Seyoum & Pradhan, 2019; Seyoum et al., 2020). We also analyzed the expression of reproduction related gene, vtg1 and found no significant change at any concentrations. Further analysis is needed to reveal the reason behind the induced reproduction.

Hypoxic conditions could occur as a result of low level of oxygen and/or disruption of heme biosynthesis due to environmental toxicants. hif1 is a transcription factor that plays a critical role in cellular response to hypoxia through several adaptive responses such as transcriptional activation of hypoxia-response element genes (Chang et al., 2019; Shiau et al., 2014). Several studies have shown that hif1 was overexpressed in response to various environmental pollutants to overcome hypoxia conditions (Eom et al., 2013; Seyoum et al., 2021). However, in the present study, we found a significant repression of hif1 upon exposure to phosmet. This indicates that D. magna may not be able to overcome hypoxic environment which led to unfavorable and stressful conditions due to phosmet toxicity. We also analyzed the expression of ftn3 and observed similar expression pattern with hif1. ftn3 encodes an iron homeostasis protein that is involved in an important storage and detoxification of excess iron in living cells (Li et al., 2012). Iron is an essential nutrient in critical biological processes such as DNA replication and ATP production. It is also used in hemoglobin biosynthesis to provide the required oxygen transportation. Excess amount of iron could induce severe damage due to reactive oxygen species production and oxidative stress (Renassia & Peyssonnaux, 2019). This suggests that it is of crucial to maintain the balance between hypoxia condition and the availability of iron in the cells. Taken together, we speculated that the downregulation of ftn3 resulted in insufficient storage of iron in the cells that could be used to avoid hypoxia.

The current study has provided significant insights into the adverse effects of phosmet on D. magna, revealing alterations in immobilization, reproduction, and gene expression profiles, alongside determining of genotoxicity and mortality. Notably, the determined LC50 values were found to be higher than those reported in existing literature (United States Environmental Protection Agency (US EPA), 2010). Additionally, our observations unveiled a hormetic response exhibited by D. magna following exposure to phosmet. While we have stated several possible explanations for these findings, it is of critical importance to emphasize the potential intraspecific variations in sensitivity to phosmet, associated with diverse genotypes and/or phases of population growth. The distinct physiological and energetic states resulting from individuals at various population growth phases, as well as genotype specificity may result in competition for limited food resources (Del Arco, Rico & van den Brink, 2015; Lyberger & Schoener, 2023; Woo, East & Salice, 2020). This competition, in turn, can impact sensitivity of Daphnids to phosmet, leading to both hormetic responses and elevated LC50 values. Nevertheless, a comprehensive understanding of these dynamics requires further in-depth analyses.

Conclusions

The excess use of anthropogenic compounds, including pesticides, results in their release into the environment and poses a serious threat to organisms and the ecosystems. Phosmet is one such pesticide which is widely used to control pests such as moths, aphids, mites, suckers and fruit flies. Elucidating the mechanisms of phosmet toxicity is of crucial to avoid the potential negative impacts of this chemical. Our findings indicate that phosmet treatment induces mortality, caused DNA damage, decreased mobility and organ movement, and altered reproduction in D. magna. Transcriptional analyses showed that phosmet can repress the expression of several genes involved in different signaling pathways including oxidative stress, detoxification, immune system, hypoxia, and iron homeostasis. This may be attributed to the observed mortality, immobility and DNA damage. In conclusion, the present study demonstrates that exposure to phosmet is lethal to D. magna in a dose- and time-dependent manner. This study also emphasizes the significance of genotoxicity assays and transcriptomic analysis in analyzing environmental pollutants. Further studies may be conducted to demonstrate the detailed effects of phosmet on reproductive system using histopathological, proteomic, and transcriptomic analyses.

Supplemental Information

Supplemental Information 1 Cycle threshold (Ct) values for qPCR.

The expression of genes involved in different signaling mechanisms were determined using qPCR analysis. After running qPCR, the raw data (Ct values) for each gene were extracted and the relative gene expressions were calculated according to the ΔΔCt method, as described by (Schmittgen & Livak, 2008). Normalization was performed using the reference gene actin (act1).

Supplemental Information 2 Supplementary Figures.

Supplemental Information 3 MIQE Checklist.

Supplemental Information 4 Mortality Data.

We would like to thank Prof. Dr. Funda Turan for her help and advices in performing comet assay.

Additional Information and Declarations

Competing Interests

Author Contributions

Data Availability

The authors declare that they have no competing interests.

Mustafa Ataş performed the experiments, analyzed the data, authored or reviewed drafts of the article, and approved the final draft.

Ceyhun Bereketoglu conceived and designed the experiments, performed the experiments, analyzed the data, prepared figures and/or tables, authored or reviewed drafts of the article, and approved the final draft.

The following information was supplied regarding data availability:

The raw data is available in the Supplemental File.

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
