# Peer review of "The toxicity assessment of phosmet on development, reproduction, and gene expression in Daphnia magna"

_PeerJ, doi:10.7717/peerj.17034_

## Round 0.1 · original submission · Major Revisions

Three recognized experts have assessed your manuscript and identified a number of issues that make the manuscript unacceptable in its present form. The most important aspects are:

- a more focused introduction and discussion
- clarification of the open questions regarding the experimental design (in particular the renewal of the medium with maintenance of stable exposure levels/concentrations and the experimental animals used: clonal line of Daphnia)
- the consideration of a possible hormetic response in the interpretation of the reproduction data

On the other hand, the reviewers have emphasized the importance of your study, so I hope that their criticism will allow you to revise the manuscript fundamentally, which is a prerequisite for the acceptance of the manuscript.

Reviewer 1 ·

Basic reporting

Dear Authors,
The results of the study are very exciting. However, some statements in the article need to be corrected. There are major errors in the article. Some of the errors mentioned are below.

In general, some of the references in the article are old. Attention should be paid to the use of current articles.

Line 57 in Abstract section; “different concentrations (0.01, 0.1, 1.10, 25, 50, 100 µM) …” must be corrected as “different concentrations (0.01, 0.1, 1, 10, 25, 50, 100 µM)”.
Such a similar mistake is made throughout the article (For example, line 186 etc.). It needs to be corrected.

In Introduction section;
Introduction must be reorganized.
Information between lines 105-115 and 116-125 is very long. Phosmet is an organophosphate, yes that's right. However, since it is organophosphate, it is not appropriate to write about pesticides and organophosphates in detail in the introduction part of the article. The pesticide whose toxicity was investigated in the article is phosmet.
There is also a similar situation in the paragraph containing lines 115 and 120. Detailed information about oxidative stress has been given. This situation is not appropriate.
Information between lines 161 and 165 should not be shown in the introduction of the article. This information should be written in the conclusion of the article.

Lines between 193 and 194 in Comet assay subtitle; “…and homogenized according to alkaline Comet assay modified from (Cavalcante et al. 2008) to obtain a single cell suspension.” should be corrected as “…and homogenized according to alkaline Comet assay modified from Cavalcante et al. (2008) to obtain a single cell suspension.”

In the article, the references written in this way should be revised.

In statistical analysis subtitle;
This statement as “For 50% lethal dose (LD50), nonlinear regression analysis was performed.” is included in the lines between lines 235-236.

However,
LD50 is the amount of a material, given all at once, which causes the death of 50% (one half) of a group of test animals (https://www.ccohs.ca/oshanswers/chemicals/ld50.html).

LC values usually refer to the concentration of a chemical in air but in environmental studies it can also mean the concentration of a chemical in water. The concentrations of the chemical in air and/or water that kills 50% of the test animals during the observation period is the LC50 values (https://www.ccohs.ca/oshanswers/chemicals/ld50.html#section-2-hdr).

Since the experiment took place in an aquatic environment, it is necessary to calculate LC50, not LD50.

This major error in the article needs to be corrected.

In Phosmet causes DNA damage in D. magna section; “…group showed 37,33% DF%.” should be corrected as “…group showed 37.33% DF%.”.

In Discussion section;
The information between lines 306 and 317 should not be included in the discussion of the article. This part has been book knowledge. Please revise it.

Please write the explanations of the superscripts ( a, b,c and *** etc.) in Table 2 below the table.

Experimental design

No comment

Validity of the findings

No comment

Additional comments

-

·

Basic reporting

1) The manuscript is clear. In general, the use of English is very good.
2) The background provided by references is acceptable, but it could be improved. There are some phenomena/processes not taken into consideration through out the manuscript (i.e., hormesis, which might explain some data behaviour)
3) Figures and tables are acceptable but they could be improved (i.e., numbre of replicas is not correct, axis titles, spacing, among others)
4) It should be of concern that authors did not take into consideration OECD guidelines for the use of solvents in toxicity tests and did not provide evidence of DMSO innocuity. Thus, some data might not be related to the effect of phosmet but to the mixture phosmet-DMSO. Analyses with daphnids seemed to be problematic. Discussion seemed to be oriented by flawly results, thus, conclusiones migth not answer the hyphotesis.

Experimental design

The manucrispt is intented to answer an interesting question about the toxicity mechanisms of phosmet. However, there are several problema with their experimental design.
Line 170 – 171: This is a very important issue that can even change results and their biological relevance. DMSO was used at 0.1% (was it volume:volume or mass:volume?) and no “solvent control” was used in further experiments. Although OECD guidelines recommend using DMSO at 0.1% (v/v) as the highest concentration, it is stated that it must demonstrated that this concentrations does not promote cytotoxic effects and does nor interfere with the assay performance. Thus, The experimental setup might not be good enough to accomplish their goal and assess the effect of phosmet but in mixture with a likely potential toxicant, the carrier solvent.

Lines 181 – 182: “Half of the exposure water was changed on alternate days”. In general, maintenance of Dpahnia magna cultures require complete renewal of the media twice a week as minimum. In the case of (sub)chronic toxicity tests, complete renewal is performed on alternate days to guarantee toxicants concentrations to remain unaltered along the duration of the experiments.

Line 235: You the term LD50 (not described before within the text). Lethal dose is not the best term when working with cladocerans as we cannot assure a dose but a concentration to which they have been exposed. Therefore, we calculate Lethal Median Concentration (LC50). The non-linear regression seems to be a good method; however, authors did not present the resultant equation and its fitness.

Validity of the findings

There are several issues to be solved.
Specific comments are included


Figure 1. n= 10? This might be a misunderstanding of the term “n” in statistics. It does not refer to the number of organisms per well but the number of replicas, in this case, n = 3.
Also, typo Daphina mangna instead of Daphnia magna
LD50s show no dispersion parameter (variance or such)

Figure 2. Do not begin a sentences abbreviating scientific names (D. magna).
Typo “<24” instead of “<24”
numerals lower than ten are written with letters; thus, “3 replicates” should be changed to “three replicates”
n = 10; the same, as there were three replicates n= 3.
The orientation of x-axis annotation can be changed from diagonal to horizontal.
Try to use the same number of decimals in all the text
More importantly, the number of offspring in the control is very unlikely for the control group. Some clutches in D. magna can reach more than 10 organisms. Therefore, these results might represent poor nourish conditions for D. magna, which were fed on Spirulina (cyanoabcteria, recognized as poor nutrient food source for cladocerans) and yeast. The validity of the test should be a higher number of offspring in optimal conditions. In 21-d experiment, at least 40 organisms as median might be acceptable rather than only 3 to 5 neonates in the controls.

Figure 3.
Try to use the same number of decimals in all the text
Do not begin a sentences abbreviating scientific names (D. magna).
No “solvent control” was included

Figure 4
Try to use the same number of decimals in all the text
Do not begin a sentences abbreviating scientific names (D. magna).
No “solvent control” was included
n = 5. How many replicas did authors include in the experiment?

Table 2
You can include a space between numbers and symbols


Discussion
Because of the experimental setup and results are inadequate, there might be several flaws within this section. Interpretation of wrong data lead to wrong conclusions. However, herein there are some specific issues:

Lines 389 – 395: It is no clear the relation of metallothioneins with the onserved effects of phosmet. Metallothioneins capture metal ions, but in phosmet there are no metal ions. Thus, what mechanisms are involved in the differential expression of these genes? Why are they altered due to phosmet exposure?

Lines 437 – 438: The hormesis phenomenon could be included and discussed. In this case, it could be better to include more concentrations to elucidate the likely mechanisms since the vtg1 gene expression was not altered.

Reviewer 3 ·

Basic reporting

The manuscript is written in clear English.

The introduction lacks some focus. The authors expand quite a lot on different aspects of pesticide pollution that do not relate to the studied pesticide, nor the organism group. For example, the authors explain how pesticides can accumulate and persist in the environment for long periods of time, even years. Even though this is true for some pesticides, it is not the case for phosmet and the majority of organophosphates, which degrade within a few days in the soil or water. The big concern regarding organophosphates in the environment is rather that they can be highly toxic at very low concentrations, particularly so for aquatic organisms. To improve the readability and relevance of the introduction, I would suggest some reformulation, particularly so in the first paragraphs, to better reflect the hazards associated with the type of pesticides used in this study, as well as it's relevance for wildlife or aquatic animals, rather than general statements focusing on humans.

I also suggest that for both the introduction and the discussion to be reformulated, so that the authors refer to more similar studies performed with the same organism using similar pesticides (other organophosphates such as chlorpyrifos or malathion) rather than compare them with studies using phosmet but performed on very different organism groups (such as vertebrates). I believe this would be more relevant to understand the obtained results, as the responses of these can be completely different. One such example is found in lines 363 – 373 - These are several examples of impacts of phosmet on male fertility, but Daphnia reproduces primarily through asexual females. Considering that toxic effects on females and males can differ, it would be relevant to also provide examples focusing on females, or at least discuss how these results could differ among sexes. Furthermore, these studies also focus on entirely different species.

The authors refer that the total number of offspring was significantly higher when exposed to 0.1 μM phosmet. It is not clear what total number of offspring refers to in this context. Does this means the sum of all offspring produced over several broods during the course of the experiment? If yes, this should be clarified. Hence, figure 1 may not be very informative. I suggest to instead show a graph with number of offspring over time. This was the result that the animals in the 0.1 μM treatment only reproduced later but produced more juveniles.

Line 242 - the title of this section mentions “developmental abnormality”, however here only results on mortality and immobilization are mentioned, so the title should be corrected.

Experimental design

Line 181 – The authors mention that half of the water was changes in alternate days in longer exposure trials. How was the concentration of phosmet maintained if part of the medium was renewed? This is very important to clarify, as variations in the final concentration that the animals were exposed to can greatly affect the results.

In the methods for the chronic exposure, please explain how the pesticide concentration was maintained. Were there several pulses of the pesticides applied throughout the experiment, and if yes when were they applied? Additionally, were the body sizes measured prior and/or during the experiment? Body size of Daphnia can be strongly related to their tolerance to different stressors, and is also very strongly correlated with fecundity. In turn, body size can also be affected by pesticide exposure, so this could be an important aspect to account for in such studies.

Validity of the findings

The study uses Daphnia magna as a model system, commonly done in toxicological studies, as they reproduce asexually and allow to work with clonal lineages. There is, however, ample evidence that there can be considerable intraspecific differences in sensitivity to stressors, associated with different genotypes. Although I do not believe that this invalidates the findings of this study, I believe it is important to acknowldge the possibility of these differences (maybe a few lines at the end of the discussion). In fact, this may be way the authors obtained LC50 values to be considerably higher than the value reported from literature.

Regarding the effects on reproduction, it is important to point out the delay in production of first brood, as well as the differences regarding total and/or average number of offspring. These results are not clearly discussed in the manuscript. First, it is important to note that brood size in Daphnia is correlated with body size – larger Daphnia produce a higher number of offspring per brood. Additionally, Daphnia body size still increases past the reaching of maturity, meaning that older individuals are larger than younger adults. This means that even though individuals exposed to phosmet showed delayed reproduction, the higher number of offspring reported could be related to a larger body size of these individuals. If data on body size of the individuals is available, it would be important to correct for this effect on number of offspring. If this data is not available, the possibility of this correlation should still be disclosed. Furthermore, considering that this effect was observed at the second lowest and sub-lethal concentration tested, maybe this could be a result of hormesis?

Additional comments

I would advise the remove the last sentences of the introduction in which the results are summarized.

Lines 311-312 and 316- “Understanding the mode of actions of these compounds including phosmet is of vital importance to take the necessary measurements by the regulatory organizations.” I assume the authors here mean the effects of organophosphates on organisms and the environment, rather than mode of action of the pesticide, which is well known. So this sentence should be reformulated to something like “understanding the impacts of these compounds” rather than mode of action.

Line 121 – a word is missing after “critical”.

---

## Round 0.2 · Minor Revisions

Reviewer 3 identified a number of points that should be considered when revising the current version of the manuscript, and I fully agree with this assessment:

- data background for fig. 2
- differences between figs. 2 and S2
- greater precision in the presentation of the reproduction test with regard to the duration and timing of neonatal production, but also to the experimental design (feed concentration, variation of phosmet concentrations over time)
- discussion on hormetic response to low-dose (!) exposures

Reviewer 3 ·

Basic reporting

I believe that the introduction has greatly improved from the previous version and is now much more focused and targeted to the topic of the research.

It is not clear to me in figure 2 if this is average number of offspring per brood released, or if this is average of all replicates for the total number of individuals that one single Daphnia produced over 21 days (all broods pooled together). Please make this more clear.
As mentioned in the previous review, I think it would greatly improve clarity of these results on reproduction if there was a graph showing brood size in the different treatments over time, so the readers can see exactly when Daphnia in each treatment started reproducing and how many juveniles were produced per brood on the different time points.

It is not clear to me what is the difference between Figure 2 and Figure S2 (in supplement), as the y-axis are the same, maybe this was a mistake? There is also no reference to figure S2 in the text. Maybe the legend of figure S2 could be made more clear, to help interpretation. Also, figure S2 seems to contradict the results, as it shows lower number of offspring for the phosmet treatments compared to the control.

The results and methods on reproduction are still confusing to me. In the discussion, the 4th paragraph starts by stating that there was a low number of offspring by the 21st day, but a few lines down the authors mention that the 0.1µM treatment started reproducing at the 24th day. I think it is important to make it clear in the methods exactly how long did this test last.

Experimental design

The authors say they performed a reproduction test for 21 days and until all animals were there. This is very confusing. Did the animals all day by day 21? Or was the test prolonged past 21 days until they were all dead?

I understand that the authors may choose to keep the data on the pesticide concentration fluctuation private, but then in the text, when referring that half of the exposure water was changed on alternate days (Line 156), add something like “and concentration of the pesticide was kept constant throughout the experiment”, to ensure the reader that there were no major variations during the 21days of the chronic exposure.

To ensure repeatability of the experiment and results, please add concentration of Spirulina and yeast given to the Daphnia cultures.

Validity of the findings

I feel like the mention of the hormesis in the discussion is rather confusing and not well explained how this is demonstrated by the results. In my previous comment, I mentioned hormesis as one possible explanation as to why individuals in the second to lowest concentration would show higher reproduction compared to the control, but I feel that the text does not clearly explore how or why the results could hint at a hormetic response. It is also important to acknowledge that it is not really possible to conclusively prove hormesis with these results, but this is clearly mentioned by the authors.

Additional comments

Line 332: it should be “may be” instead of “maybe”

Line 337: A hormetic response refers to an improve in fitness to response to a low-dose exposure, not high dose as mentioned in the discussion, so please revise this sentence

---

## Round 0.3 · accepted · Accept

Thank you for the revision of the manuscript. I hereby certify that you have adequately taken into account the reviewers' comments, as I have checked by my own assessment of your revised manuscript. Based on my assessment as an Academic Editor, your manuscript is now ready for publication.